# Modulation of *Staphylococcus aureus* gene expression during proliferation in platelet concentrates with focus on virulence and platelet functionality

Basit Yousuf [1,2], Roya Pasha[1], Nicolas Pineault[1,2], Sandra Ramirez-Arcos[1,2]*

1 Medical Affairs and Innovation, Canadian Blood Services, Ottawa, Canada, 2 Department of Biochemistry, Microbiology and Immunology, University of Ottawa, Ottawa, Canada

* sandra.ramirez@blood.ca

**Data Availability Statement:** The RNA-seq data is accessible through the NCBI Gene Expression Omnibus (GEO) under accession number GSE241582.

## Abstract

*Staphylococcus aureus* is a well-documented bacterial contaminant in platelet concentrates (PCs), a blood component used to treat patients with platelet deficiencies. This bacterium can evade routine PC culture screening and cause septic transfusion reactions. Here, we investigated the gene expression modulation within the PC niche versus trypticase soy media (TSB) of *S. aureus* CBS2016-05, a strain isolated from a septic reaction, in comparison to PS/BAC/317/16/W, a strain identified during PC screening. RNA-seq analysis revealed upregulation of the capsule biosynthesis operon (*capA-H*), surface adhesion factors (*sasADF*), clumping factor A (*clfA*), protein A (*spa*), and anaerobic metabolism genes (*pflAB, nrdDG*) in CBS2016-05 when grown in PCs versus TSB, implying its enhanced pathogenicity in this milieu, in contrast to the PS/BAC/317/16/W strain. Furthermore, we investigated the impact of *S. aureus* CBS2016-05 on platelet functionality in spiked PCs versus non-spiked PC units. Flow cytometry analyses revealed a significant decrease in glycoprotein (GP) IIb (CD41) and GPIbα (CD42b) expression, alongside increased P-selectin (CD62P) and phosphatidylserine (annexin V) expression in spiked PCs compared to non-spiked PCs ($p = 0.01$). Moreover, spiked PCs exhibited a drastic reduction in MitoTrack Red FM and Calcein AM positive platelets (87.3% vs. 29.4%, $p = 0.0001$ and 95.4% vs. 24.7%, $p = 0.0001$) in a bacterial cell density manner. These results indicated that *S. aureus* CBS2016-05 triggers platelet activation and apoptosis, and compromises mitochondrial functionality and platelet viability, in contaminated PCs. Furthermore, this study enhanced our understanding of the effects of platelet-bacteria interactions in the unique PC niche, highlighting *S. aureus* increased pathogenicity and deleterious effect on platelet functionality in a strain specific manner. Our novel insights serve as a platform to improve PC transfusion safety.

**Funding:** The project was funded by Canadian Blood Services (intramural grant awarded to Sandra Ramirez-Arcos) and Health Canada.

**Competing interests:** The authors have declared that no competing interests exist.

## Introduction

Platelet concentrates (PCs) play a critical role in transfusion therapy for patients with thrombocytopenia or impaired platelet functionality. Ensuring the safety of PCs is of utmost importance, considering the millions of PC transfusions conducted annually in the United States alone. PCs are stored in gas-permeable plastic containers under agitation at temperatures of 20–24°C for up to 7 days. These storage conditions make PCs highly vulnerable to proliferation of contaminant bacteria introduced during blood collection [1]. Among the microbiological causes of post-transfusion severe reactions, bacterial contamination of PCs and subsequent transfusion-mediated infections and fatalities pose significant clinical challenges [2].

*Staphylococcus aureus*, an opportunistic pathogen, is a major reported bacterial contaminant in PCs [1, 3–5]. It enters donated blood during venipuncture at the time of blood collection and can evade detection with automated culture methods, leading to septic transfusion reactions as observed in cases reported in Canada, the US, and the UK [4, 6–8]. In healthy individuals, *S. aureus* colonizes human anterior nares, with approximately 20–30% of asymptomatic population persistently carrying the bacterium and 20–60% carrying it transiently [9, 10]. Persistent carriers have a higher likelihood of *S. aureus* colonization on the skin, which can subsequently contaminate PCs if they donate blood.

*S. aureus* is well-known for causing nosocomial and community-acquired infections and is notorious for its resistance mechanisms against antibiotics and antimicrobial peptides [11]. Formation of surface attached aggregates known as biofilms, and other virulent factors including production of enzymes (e.g., coagulases, proteases) and exotoxins, significantly contribute to *S. aureus* virulence and persistence [12]. *S. aureus* isolates exhibit strongly enhanced biofilm formation within the PC environment, and even the biofilm-negative strains turn into biofilm-positive entities [13, 14]. Staphylococcal biofilm matrix usually contains the polysaccharide intercellular adhesin (PIA) encoded by the *icaADBC* operon [15–17]; however; in PCs, the biofilm matrix of the common PC contaminant *Staphylococcus epidermidis* consist mainly of proteins and extracellular DNA (eDNA) [13]. Various colonization and intercellular adhesion factors, including microbial surface components recognizing adhesive matrix molecules (MSCRAMMs), are involved in the process of biofilm formation by staphylococci [18]. MSCRAMMs such as fibronectin-binding proteins (FnbA, B), serine-aspartate repeat family proteins (SdrC, SdrD, and SdrE), and clumping factors (ClfA and ClfB), interact with host matrix factors like fibronectin and fibrinogen, while host-derived fibrin also contributes to the biofilm matrix of *S. aureus* [19, 20]. Biofilms provide protection against the host immune system and confer enhanced resistance to antibiotics and antimicrobial peptides [21]. *S. aureus* utilizes factors such as ClfA, protein A, and Fnb to interact with platelets, resulting in platelet activation and aggregation [22, 23]. However, neither of these interactions has been explored in the context of PCs or bacteria-mediated transfusion reactions.

Despite the critical role of platelet-bacteria interactions in PC contamination, molecular modifications of *S. aureus* and impact on platelet functionality remain poorly understood. To address this, we conducted RNA-seq analysis of *S. aureus* during its interaction with PCs to uncover potential factors linked to immune evasion and missed detection during PC screening. Furthermore, we have recently reported that *S. aureus* induces metabolic changes in contaminated PCs [24], which was further explored herein with flow cytometry assessing platelet functionality. By integrating RNA-seq analysis and flow cytometry, we aim to comprehensively understand underlying molecular mechanisms of *S. aureus*-platelet interactions. This exploration will enhance our understanding of PC contamination and guide targeted interventions for safer PC transfusions.

## Materials and methods

### Platelet concentrates preparation and ethics approval

PCs were manufactured from whole blood donations using the buffy coat pooling method with donor consent obtained in writing and ethical approval granted by the Canadian Blood Services Research Ethical Board (REB 2015.024 AND 2017.033). Blood collection and PC preparation was done by the Canadian Blood Services Blood for Research Facility (Blood4Research, Vancouver, BC, Canada) in agreement with Canadian Blood Services procedures. PC pools were suspended in 100% plasma and shipped to the Canadian Blood Services Microbiology laboratory in Ottawa, Ontario, Canada, where they were screened for bacterial contamination upon arrival with the BACT/ALERT 3D system following standard procedures [25].

### Bacterial strains, plasmids, and growth conditions

Two *S. aureus* strains namely CBS2016-05 and PS/BAC/317/16/W isolated from contaminated PCs in Canada and England, respectively, were selected for this study. CBS2016-05 was isolated post-transfusion after causing a septic reaction in an elderly leukemia patient [7] whereas the PS/BAC/317/16/W strain was detected during PC screening with the automated BACT/ALERT culture system [4]. The *S. aureus* strains were routinely cultured on Trypticase Soy Agar (TSA) for colony isolation or in Trypticase Soy Broth (TSB) without or with 0.5% glucose (TSBg) and incubated with agitation (20–24˚C under agitation for 6 days) or static conditions at 37˚C for 24 h. In PCs, the strains were grown aerobically at 20–24˚C under agitation for 6 days.

### *S. aureus* RNA isolation, library construction and sequencing

For RNA isolation, the CBS2016-05 and PS/BAC/317/16/W strains were spiked into three independent PC pools and TSB cultures, with an initial concentration of approximately ~4E+06 colony forming units (CFU)/PC unit. The spiked cultures were allowed to grow to the stationary phase at 20–24˚C under agitation. Subsequently, the cells were pelleted at 4˚C and subjected to total RNA extraction using the miRNeasy Mini Kit (Qiagen), following the manufacturer's instructions. To eliminate genomic DNA, the RNA samples from three independent biological replicates were treated with TURBO™ DNase AmbionTM (Thermo Fisher Scientific). Furthermore, the RNA samples from spiked PCs underwent an additional treatment using the MICROBEnrich™ kit (Ambion) to remove eukaryotic RNA. RNA sequencing was performed at the Ottawa Hospital Research Institute (OHRI) sequencing facility. Briefly, quality and quantity of the RNA samples were assessed using a Biodrop µLITE and Fragment Analyzer™, revealing an RNA Quality Number (RQN) ranging between 8.7 and 10 for all samples. Subsequently, cDNA libraries were generated using Illumina® Stranded Total RNA Prep, Ligation with Ribo-Zero Plus (Illumina 20040525). Quantification of the libraries was performed with the Qubit Double Stranded DNA HS kit (Thermo Q32854) and ran on the AATI Fragment Analyzer to verify the size distribution. The libraries were normalized, pooled, and diluted as required to achieve acceptable cluster density on the NextSeq 500 sequencer (Illumina SY-414-1001). The library pool then underwent 75 Cycle High Output (Illumina 20024906).

### Transcriptome assembly and differential gene expression analyses

Genome sequences of *S. aureus* CBS2016-05 and PS/BAC/317/16/W have been annotated by our laboratory and uploaded into NCBI [26, 27]. The reference genome sequences were used for transcriptome assembly. Sequence quality control analysis was performed using FastQC

and fastp to assess the quality of the reads. Transcript quantification was conducted using Salmon and the reads were aligned to the genome using bowtie2. For differential expression analysis, the transcript quantification data for all 12 samples were imported into R. In the subsequent stages of the analysis, genes encoding rRNA and tRNA were filtered out from the read count matrix, and non-expressed genes (genes with fewer than two replicates with five or more assigned reads) were excluded. DESeq2 was employed to estimate size factors for count scaling based on library size and to calculate dispersion parameters to assess the deviation of expression variance from the mean across the dataset [28]. The DESeq 'rlog()' function was applied to perform a regularized log transformation of the count values, which were then used in DESeq's plotPCA() function to generate a PCA plot visualizing the clustering of the replicates. Hierarchical clustering was conducted using the rlog-transformed values to depict similarities between samples. Following the normalization procedure, differentially expressed gene (DEG) analysis was carried out between the PC and TSB conditions for each bacterial strain. The 'lfcShrink()' function in DESeq2 was employed for this analysis, which calculates the log2 fold change between the conditions while shrinking the value in cases of high uncertainty in the estimated fold change, often arising from low read counts assigned to the gene. The analysis also generated p-values indicating the probability of true differential expression between the conditions, and FDR/q-values were calculated to correct the p-values for multiple testing using the Benjamini-Hochberg approach. The RNA-seq data is accessible through the NCBI Gene Expression Omnibus (GEO) under accession number GSE241582. Gene ontology (GO) functional enrichment analysis of these DEGs was conducted using the ShinyGO enrichment tool.

## Validation of RNA-seq data using quantitative reverse transcription PCR (RT-qPCR)

For the validation of RNA-seq data, qRT-qPCR was performed on eleven randomly selected DEGs in triplicate for all the three independent replicates to corroborate expression pattern of RNA-seq data sets. Complementary DNA (cDNA) synthesis was performed with 1 µg of DNase treated total RNA using the iScript cDNA synthesis kit (Bio-Rad) following manufacturer's instructions and using the following protocol: Priming 5 min at 25°C; Reverse transcription 20 min at 46°C; RT inactivation 1 min at 95°C. The qRT-PCR reaction was carried out in a CFX96 Thermal cycler (Bio-Rad) using iQ SYBR Green super mix (Bio-Rad) as recommended by the manufacturer. Oligonucleotide primers were designed using the PrimerQuest tool at IDT website and efficiency of the primers was determined by the dilution method (100 ng– 0.01 ng) using following protocol: 3 min at 95°C; 40 cycles (10 sec at 95°C; 25 sec at 57°C); 10 sec at 95°C; 5 sec at 65°C. Melt curves were produced to assess the specificity and efficiency of the primers. DNA gyrase A (*gyrA*) was used as an internal control to normalize the target gene mRNA expression. Fold change in gene expression between TSB and PC samples were calculated using cycle threshold (Ct) values of each gene and the $2^{-\Delta\Delta Ct}$ method [29]. The sequences of the primers used in these procedures are shown in **S1 Table**.

## *S. aureus* growth curves

Growth curves were performed to compare the growth dynamics of *S. aureus* CBS2016-05 and PS/BAC/317/16/W at different time intervals in TSB and PCs. The bacterial strains with initial inocula ($OD_{600}$ = 0.002), which corresponds to ~4E+06 colony forming units (CFU)/mL, were independently inoculated in TSB and PC cultures, and allowed to grow in PC conditions. Samples were withdrawn at the different time points for plating on TSA to determine bacteria concentration (CFU/mL).

## Semi-quantitative biofilm assay

*S. aureus* strains were grown in TSB overnight at 37˚C with agitation. An inoculum was added to TSBg and PCs to adjust OD600 to 0.1 and then, 3-mL culture suspensions were dispensed into 6-well polystyrene plates (Falcon, Corning Inc., Durham, NC) and incubated for 24, 48, 72, and 144 h at 37˚C or 20–24˚C under agitation. After the specified incubation times, supernatants were discarded, and adhered cells (biofilms) were washed three times using 1X PBS and air dried. Biofilms were then stained with Gram crystal violet dye (BD Biosciences, MD, USA) for 30 min and subsequently washed 3 times with 1X PBS. De-staining was done with 80:20 ethanol: acetone solution for 15 min and absorbance was measured at 492 nm in microplate reader. Biofilm quantification was performed from absorbance value subtracting baseline reading obtained with uninoculated TSB and PCs following the recommendations for staphylococci [30].

## Flow cytometry analyses

Three buffy coat PC units, each divided into two units were utilized for flow cytometry analyses, from each pair of PCs, one split unit was spiked with *S. aureus* CBS 2016–05 (at a concentration of 1E+06 CFU/bag) while the second split PC served as control. Platelet counts were determined using the Sysmex pocH-100i™ Automated Haematology Analyser (Sysmex Corporation, Kobe, Hyogo, Japan). The platelet samples were diluted to a concentration of 10–40 x $10^6$ platelets/mL in either phosphate-buffered saline (PBS) or Annexin binding buffer. In the subsequent steps, a range of specific staining procedures were carried out to discern various platelet characteristics. A volume of fifty µL from the diluted platelet samples were stained using platelet-specific Allophycocyanin (APC)-conjugated CD41a (GPIIb) antibody (5 µl) (BD Bioscience), phycoerythrin (PE)- conjugated CD62P (P-selectin) antibody (10 µl) (BD Bioscience) as a marker of platelet activation, and fluorescent dye MitoTrack™ Red FM (ABP Biosciences) (2 µL), employed to demonstrate platelet mitochondrial activity. In another staining, Fluorescein-5-isothiocyanate (FITC) conjugated CD42b (GPIbα) antibody (5 µL) (BD Bioscience) was utilized as another platelet marker, and Alexa 488 conjugated Annexin V (1.5 µL) (Thermo Fisher Scientific) was employed to label phosphatidylserine sites on the platelet membrane surface. The staining reactions were brought to a final volume of 100 µL, adjusted with the appropriate buffer, and subsequently stored in a light-protected environment at room temperature (RT) for a duration of 20 minutes. Finally, 400 µL of PBS or Annexin binding buffer was incorporated into each sample prior to analysis on an Attune acoustic focusing cytometer equipped with 488 and 637 nm lasers (Life Technologies, ThermoFisher Scientific, Waltham, MA, USA). Additionally, separate staining using CD41a-APC was performed, involving a 20-minute incubation at RT in darkness, followed by adjusting the volume to 500 µL and addition of Calcein AM (2 uL) (Thermo Fisher Scientific) and further incubation in darkness at RT for 45 minutes and subsequent analysis on flow cytometer. MitoTrack Red FM and Calcein AM diluted solutions were freshly prepared in PBS and DMSO, respectively.

## Statistical analyses

Statistical significance levels were determined using 2way ANOVA followed by the Tukey tests using Prism software and Excel software was used to perform two-tailed T-test. Statistical analysis of the RNA-seq data was done in R. Values were expressed as mean ± SE and a *p*-value of <0.05 was considered statistically significant.

## Results and discussion

In modern medicine, PCs hold immense importance, especially in transfusion medicine and hematology, with millions of PC units transfused globally each year. Ensuring PC safety becomes paramount due to potential risks associated with contamination by bacterial pathogens like *S. aureus*. This bacterium can evade immune defenses mounted by platelets, survive and proliferate in PCs, with some strains evading routine PC screening causing septic transfusion reactions which may result in fatalities [7, 8, 31, 32]. Herein, we studied molecular modulations of *S. aureus* and its impact on platelet functionality during PC storage.

### Time course growth curve and comparative RNA-seq revealed that *S. aureus* CBS2016-05 and PS/BAC/317/16/W behave differently in PCs

In this study, we compared the growth dynamics of *S. aureus* strains CBS2016-05 (missed during PC screening) and PS/BAC/317/16/W (detected during PC screening) in PCs and TSB incubated under PC storage conditions (20–24˚C under agitation) until they reached early stationary phase. Results showed that CBS2016-05 strain is a slow growing strain in TSB and reached early stationary phase at 96 hr whereas PS/BAC/317/16/W reached stationary phase at 40 hr. Conversely, in PCs, stationary phase was reached at 144h by both the strains and their growth was significantly slower than their growth in TSB conditions (padj = 0.0002) (**Fig 1A and 1B**).

These data demonstrate that the PC storage environment is challenging for *S. aureus* proliferation likely due to low nutrient availability and immune factors present in plasma and released by platelets. Accumulating evidence have substantiated the potent ability of platelets to defend against *S. aureus* attacks, exhibiting either direct killing of *S. aureus* or their phagocytosis by macrophages, leading to intracellular elimination [33, 34]. Upon contact with *S. aureus*, platelets undergo activation, aggregation, or de-granulation, subsequently releasing a repertoire of antimicrobial compounds collectively referred to as platelet microbicidal proteins (PMPs) [35–37]. However, many *S. aureus* strains survive and proliferate within PCs indicating that *S. aureus* has the ability to activate immune evasion mechanisms against immune factors [6–8].

Comparative transcriptome profiling performed at early stationary phase unveiled 358 and 315 differentially expressed genes (log2FoldChange ≥2 or ≤-2, p<0.05,) [28] in *S. aureus*

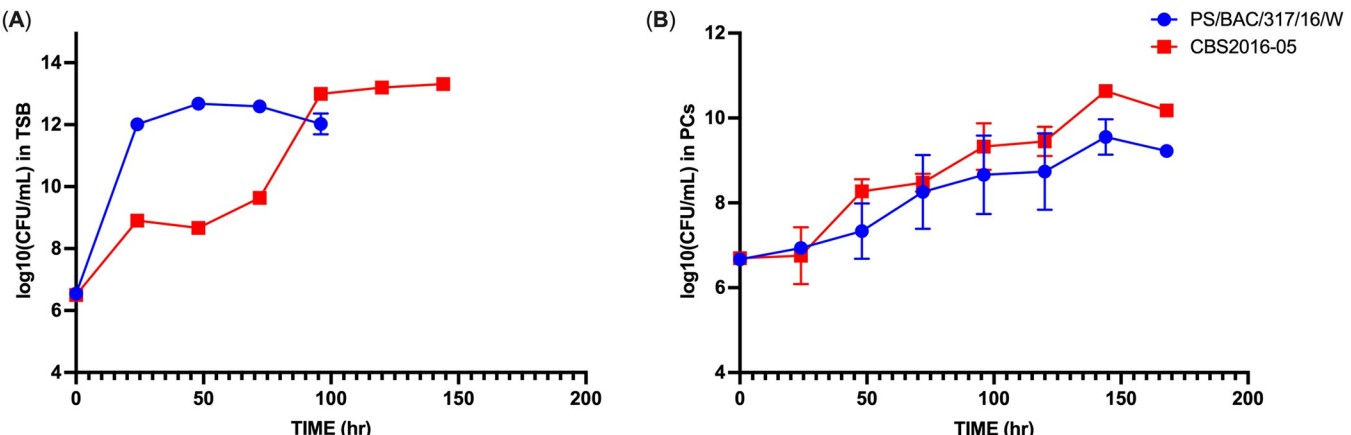

**Fig 1. Decreased growth of *S. aureus* CBS2016-05 and PS/BAC/317/16/W during PC storage in comparison to trypticase soy broth (TSB).** Time-course growth curves of CBS2016-05 and PS/BAC/317/16/W in (**A**) TSB and (**B**) PCs. The samples were inoculated in three independent replicates with an initial concentration of ~4E+06 CFU/mL. The curves were generated using log10 CFU/ml values calculated at 0, 24, 48, 72, 96 and 144 hr.

CBS2016-05 and PS/BAC/317/16/W, respectively (S2 Table). Principal component analysis (PCA) of the DEGs revealed samples spiked in TSB tend to cluster together whereas the samples in PCs show more dispersion indicating more variability (Fig 2A and 2B). Dispersion of samples in PCs likely reflect the characteristic variability inherent to each PC donor [38] as not all three PC units cluster together (Fig 2A and 2B). Amongst the DEGs, we found 181 and 133 genes with significant upregulation (≥2-fold) and 178, 181 genes with significant downregulation (≤-2-fold) in CBS2016-05 and PS/BAC/317/16/W strains, respectively (Fig 2C and 2D). Gene ontology (GO) analysis of these DEGs revealed enrichment of amino acid, organic acid, and cellular metabolic processes in both strains, except for pathogenesis, which was enriched solely in the CBS2016-05 strain (S1A and S1B Fig). *S. aureus* possesses robust metabolic capabilities, enabling it to persist and colonize challenging environments through the production of a diverse array of virulence factors, immune evasion mechanisms, and various proteins and metabolites crucial for its survival. Additionally, *S. aureus* adeptly employs different carbon and nitrogen sources for its successful colonization within specific niches [39]. Given this, we focused on analyzing key pathways that underpin the survival, persistence, and proliferation of *S. aureus* in PCs.

## Capsule biosynthesis gene expression by *S. aureus* CBS2016-05 is highly enhanced in PCs

To elucidate the trend of *S. aureus* gene expression, we compared major *S. aureus* pathways modulated between CBS2016-05 and PS/BAC/317/16/W in PCs versus TSB (Table 1). A

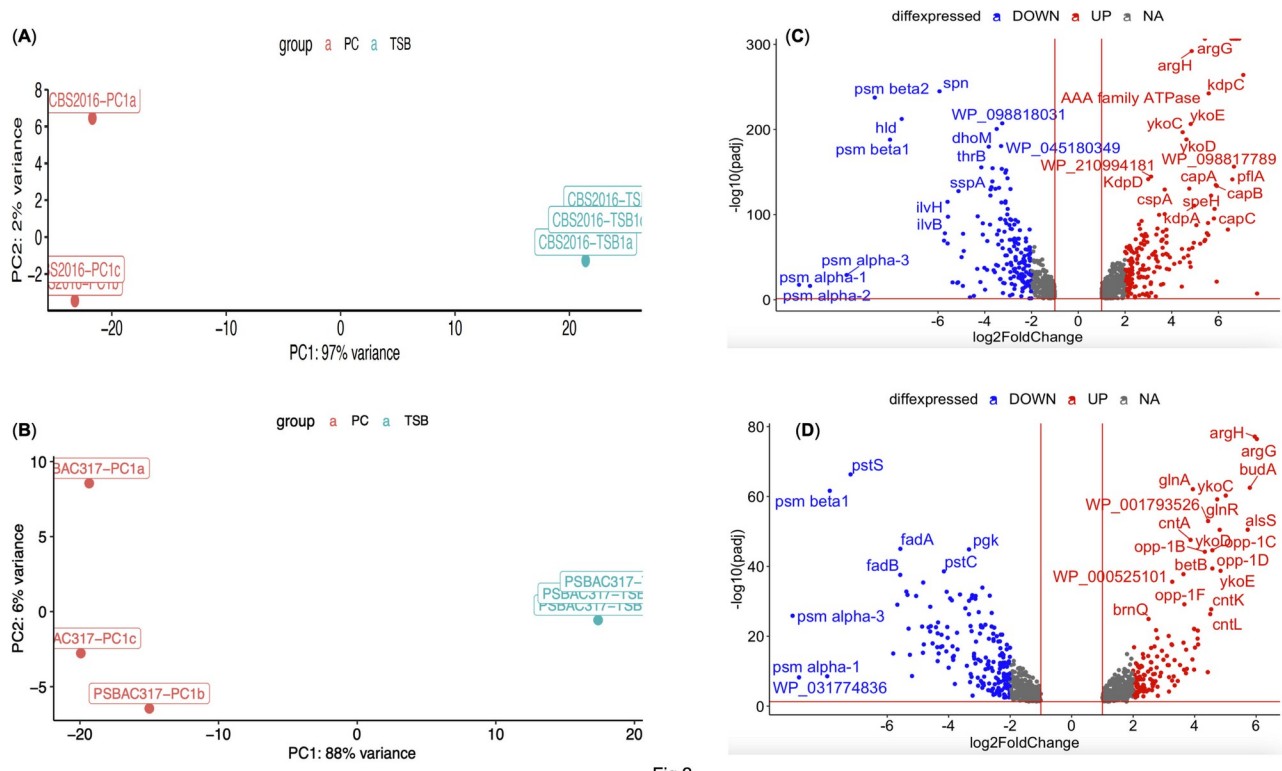

Fig 2

**Fig 2. The PC environment induces expression of key virulence and immune evasion gene pathways in CBS2016-05.** PCA plots (PCs vs TSB) of *S. aureus* (**A**) CBS2016-05 and (**B**) PS/BA/317/16/W strains showing the overall structure of the transcriptomic data. (**C**) Volcano plot illustrating differentially expressed genes (DEGs) (log fold-change ≥ 2 or ≤ -2, *p*<0.05) by *S. aureus* CBS2016-05 in PCs versus TSB. (**D**) Volcano plot illustrating differentially expressed genes (DEGs) (log fold-change ≥ 2 or ≤ -2, *p*<0.05) by *S. aureus* PS/BA/317/16/W in PCs versus TSB. Scattered blue and red dots represent down-regulated and up-regulated DEGs, respectively, with significant differences indicated by -log(10) p-value < 0.05.

notable observation was the 16-gene operon required for capsule biosynthesis with genes (*capABCD*) that were strongly upregulated in CBS2016-05 in PCs with 5.8-fold upregulation whereas in PS/BAC/317/16/W, these genes were not differentially expressed, implying enhanced capsule formation in CBS2016-05 when grown in PCs. Genes encoding for *saeRS*, the two-component system (TCS) which negatively regulates capsule biosynthesis [40], were preferentially repressed in CBS2016-05 (-3-fold) (**Table 1**).

Capsular polysaccharides (CPs) are decorated on the peptidoglycan layer along with wall teichoic acid (WTA) and have a vital function in virulence, impeding phagocytosis as well as equipping bacteria to evade immune defenses thus enabling bacterial persistence [41, 42]. Two types of serotypes of CPs are produced by *S. aureus* viz. CP 5 and 8 [43, 44] which are known

**Table 1. List of selected highly differentially expressed genes from *S. aureus* CBS2016-05 and PS/BAC/317/16/W spiked-PCs vs spiked-TSB.**

| Proteins | Genes | CBS2016-05 log2fold | PS/BAC/317/16/W log2fold | Pathways |
|---|---|---|---|---|
| accessory gene regulator AgrB | *agrB* | -2.00 | -1.79 | Agr quorum sensing/ surface proteins |
| cyclic lactone autoinducer peptide | *agrD* | -1.99 | -2.02 | |
| GHKL domain-containing protein | *agrC* | -2.00 | -1.82 | |
| response regulator transcription factor | *agrA* | -2.11 | -1.63 | |
| bi-component gamma-hemolysin HlgAB/HlgCB | *hlgB* | -3.21 | -2.44 | |
| bi-component gamma-hemolysin HlgCB subunit C | *hlgC* | -2.82 | -2.50 | |
| bi-component gamma-hemolysin HlgAB subunit A | *hlgA* | -2.92 | -5.26 | |
| delta-hemolysin | *hld* | -7.55 | -2.19 | |
| alpha-hemolysin | *hyl* | 1.22 | -1.68 | |
| phenol-soluble modulin PSM-alpha-1 | *PSM-alpha-1* | -11.93 | -7.95 | |
| phenol-soluble modulin PSM-alpha-2 | *PSM-alpha-2* | -11.48 | | |
| phenol-soluble modulin PSM-alpha-3 | *PSM-alpha-3* | -9.91 | -9.08 | |
| phenol-soluble modulin PSM-alpha-4 | *PSM-alpha-4* | -4.63 | | |
| beta-class phenol-soluble modulin-1 | *psm beta1* | -8.71 | -7.87 | |
| beta-class phenol-soluble modulin-2 | *psm beta2* | -8.06 | | |
| Glu-specific serine endopeptidase | *sspA* | -5.13 | 2.00 | |
| cysteine protease staphopain B | *sspB* | -4.09 | | |
| staphostatin B | *sspC* | -4.09 | | |
| cell-wall-anchored protein | *sasF* | 1.70 | | |
| LPXTG cell wall anchor domain-containing protein | *sasD* | 2.82 | -3.87 | |
| serine-rich repeat glycoprotein adhesin SasA | *sasA* | 1.96 | | |
| fibrinogen-binding adhesin SdrG C-terminal | *sdrG* | | 1.34 | |
| autolysin/adhesin | *aaa* | -2.45 | 2.79 | |
| MSCRAMM family adhesin clumping factor | *clfA* | 1.55 | -1.39 | |
| MSCRAMM family adhesin clumping factor | *clfB* | 2.24 | 2.58 | |
| protein VraX | *vraX* | 3.02 | | Immune evasion |
| staphylococcal protein A | *spa* | 5.80 | -1.52 | |
| myeloperoxidase inhibitor SPIN | *spn* | -5.94 | -3.22 | |
| complement inhibitor SCIN-A | *scn* | -3.04 | -2.43 | |
| extracellular adherence protein Eap/Map | *eap* | -3.57 | | |
| MAP domain-containing protein | *map* | -2.80 | | |

*(Continued)*

**Table 1.** (Continued)

| Proteins | Genes | CBS2016-05 log2fold | PS/BAC/317/16/W log2fold | Pathways |
|---|---|---|---|---|
| staphylocoagulase | *coa* | -3.89 | | Other virulence factors |
| von Willebrand factor binding protein Vwb | *vwb* | 1.41 | | |
| YSIRK domain-containing triacylglycerol lipase | *lip1* | -2.24 | -5.34 | |
| YSIRK domain-containing triacylglycerol lipase | *lip2/geh* | 2.28 | -1.22 | |
| phosphatidylinositol-specific phospholipase C | *plc* | | 2.61 | |
| type 8 capsular polysaccharide synthesis protein | *cap8P* | 2.46 | 1.55 | Capsule Biosynthesis |
| type 8 capsular polysaccharide synthesis protein | *cap8O* | 4.11 | 1.50 | |
| capsular polysaccharide type 5/8 biosynthesis epimerase | *capN* | 4.35 | 1.80 | |
| type 8 capsular polysaccharide synthesis protein | *cap8M* | 4.52 | 1.84 | |
| type 8 capsular polysaccharide synthesis protein | *cap8L* | 4.70 | 2.04 | |
| capsular biosynthesis protein | *cap8K* | 4.89 | 2.08 | |
| O-antigen ligase family protein | *cap8J* | 4.74 | 2.03 | |
| glycosyltransferase | *cap8I* | 4.94 | 2.13 | |
| antibiotic acetyltransferase | *cap8H* | 5.22 | 1.81 | |
| type 8 capsular polysaccharide synthesis protein | *cap8G* | 5.40 | 1.64 | |
| type 8 capsular polysaccharide synthesis protein | *cap8F* | 5.50 | 1.50 | |
| type 8 capsular polysaccharide synthesis protein | *cap8E* | 5.43 | 1.23 | |
| type 8 capsular polysaccharide synthesis protein | *cap8D* | 5.56 | | |
| type 8 capsular polysaccharide synthesis protein | *cap8C* | 5.83 | | |
| type 8 capsular polysaccharide synthesis protein | *cap8B* | 5.93 | | |
| capsular polysaccharide type 5/8 biosynthesis | *capA* | 5.89 | | |
| response regulator transcription factor | *saeR* | -3.15 | -1.44 | Regulatory proteins |
| two-component system sensor histidine kinase SaeS | *saeS* | -3.33 | -1.31 | |
| response regulator transcription factor | *kdpE* | 2.24 | 1.86 | Potassium transport |
| sensor histidine kinase KdpD | *kdpD* | 2.57 | 2.01 | |
| potassium-transporting ATPase subunit A | *kdpA* | 4.97 | 2.62 | |
| potassium-transporting ATPase subunit KdpB | *kdpB* | 3.77 | 1.38 | |
| K(+)-transporting ATPase subunit C | *kdpC* | 3.82 | 1.21 | |
| K(+)-transporting ATPase subunit C | *kdpC* | 7.06 | | |
| K(+)-transporting ATPase subunit B | *kdpB* | 6.79 | | |
| potassium-transporting ATPase subunit A | *kdpA* | 6.55 | | |
| sensor histidine kinase KdpD | *kdpD* | 2.99 | | |
| response regulator transcription factor | *kdpE* | 2.71 | | |
| pyruvate formate lyase-activating protein | *pflA* | 6.59 | 1.95 | Anaerobic/aerobic metabolism |
| formate C-acetyltransferase | *pflB* | 4.44 | | |
| acetolactate synthase AlsS | *alsS* | 6.89 | 5.73 | |
| acetolactate decarboxylase | *budA* | 6.76 | 5.79 | |
| anaerobic ribonucleoside-triphosphate reductase | *nrdD* | 3.12 | 1.09 | |
| anaerobic ribonucleoside-triphosphate reductase activating protein | *nrdG* | 5.05 | 2.65 | |
| class 1b ribonucleoside-diphosphate reductase subunit beta | *nrdF* | | 2.48 | |
| class 1b ribonucleoside-diphosphate reductase subunit alpha | *nrdE* | | 2.38 | |
| class Ib ribonucleoside-diphosphate reductase assembly flavoprotein NrdI | *nrdI* | | 2.32 | |

*(Continued)*

**Table 1.** (Continued)

| Proteins | Genes | CBS2016-05 log2fold | PS/BAC/317/16/W log2fold | Pathways |
|---|---|---|---|---|
| threonine ammonia-lyase IlvA | *ilvA* | -4.94 | 1.22 | Amino Acid metabolism |
| 3-isopropylmalate dehydratase small subunit | *leuD* | -5.14 | 1.08 | |
| 3-isopropylmalate dehydratase large subunit | *leuC* | -5.18 | | |
| 3-isopropylmalate dehydrogenase | *leuB* | -5.38 | | |
| 2-isopropylmalate synthase | | -5.58 | | |
| ketol-acid reductoisomerase | *ilvC* | -5.74 | | |
| ACT domain-containing protein | *ilvH* | -5.59 | 1.87 | |
| biosynthetic-type acetolactate synthase large subunit | *ilvB* | -5.57 | 1.94 | |
| dihydroxy-acid dehydratase | *ilvD* | -4.93 | 2.25 | |
| branched-chain amino acid transport system II carrier protein | *brnQ* | 1.68 | 2.50 | |
| betaine-aldehyde dehydrogenase | *betB* | 6.74 | 3.64 | |
| choline dehydrogenase | *betA* | 3.46 | 2.75 | |
| superoxide dismutase | | 3.02 | | |
| superantigen-like protein SSL14 | *SSL14* | 6.66 | | Superantigens |
| superantigen-like protein SSL13 | *SSL13* | 5.68 | | |
| superantigen-like protein SSL12 | *SSL12* | 4.75 | | |
| superantigen-like protein SSL10 | *SSL10* | 1.41 | | |
| staphylococcal enterotoxin type H | *seh* | 2.30 | | |
| exotoxin | | 3.58 | | |
| oleate hydratase | *ohyA* | 5.55 | | |

to be involved in invasive diseases. Upregulation of the *cap* operon by CBS2016-05 in PCs highlights how virulence is triggered by the PC storage challenging environment, which likely contributed to this strain being involved in a septic transfusion reaction [7].

## Suppression in expression of *agr* quorum sensing pathway genes coupled with upregulation of biofilm-related genes

Another important observation from RNA-seq data was the complete repression in transcription of the accessory gene regulator (*agr*) quorum sensing pathway. Agr is the master regulator of *S. aureus* virulence and controls the production of numerous virulence factors through *agrA*, and *RNAIII*, which positively regulates the production of phenol soluble modulins (PSM) and proteases, and negatively controls the expression of surface adhesins. We observed strong repression of genes encoding for *psm-alpha1-4*, *psm-beta1,2* (up to -11.9-fold) and *hld* (-7.5-fold) in CBS2016-05 whereas lesser repression of genes *psm-alpha1,3* (up to -9-fold); *psm-beta1* (-7-fold); and *hld* (-2.1-fold) in PS/BAC/317/16/W (**Table 1**). Surface adhesins like *clfA*, *clfB*, and *sasADF* which promote biofilm formation were upregulated (~2-fold) thus implying enhanced biofilm formation in PCs by CBS2016-05 [7, 45]. Increased biofilm formation was observed in PCs compared to TSB as recently demonstrated [14]. Downregulation of *agr* and *psm* genes have been previously implicated in enhanced biofilm biosynthesis, colonization, and persistence [46, 47]. The *icaBDCA* operon which is associated with the formation of PIA-mediated biofilm matrix, did not show significant differential expression. The lack of upregulation in the *ica* genes, despite other observations pointing towards an increased biofilm presence, can be ascribed to their function as adhesion factors during the initial phases of biofilm formation [48]. Furthermore, our studies have previously highlighted that *Staphylococcus* species predominantly form protein and eDNA-mediated biofilms in PCs [13, 49].

## Differential gene expression of immune evasion factors in PCs

Major immune evasion factors such as myeloperoxidase inhibitor SPIN (*spn*), complement inhibitor SCIN-A (*scn*), extracellular adherence protein Eap/Map (*eap*), and MAP domain-containing protein (*map*) were significantly downregulated in both strains grown in PCs compared to TSB. In contrast, the gene encoding for staphylococcal protein A (*spa*) was upregulated in CBS2016-05 (5.8-fold) and downregulated in PS/BAC/317/16/W (-1.5-fold) (**Table 1**). Protein A has been implicated in immune evasion and sepsis [50–52]. It is known to inhibit the host's humoral immune response by blocking opsonophagocytosis and inducing apoptosis [53]. Moreover, *vraX* was upregulated only in CBS2016-05 strain and the VraX protein was recently reported to bind to C1q for the inhibition of classical complement pathway consequently promoting *S. aureus* survival and pathogenesis [54].

## Enriched anaerobic metabolic activity of CBS2016-05 grown in PCs

Unexpectedly, we also observed significant upregulation of genes with important physiological roles in anaerobic metabolism such as pyruvate formate lyase (*pflA*) and formate acetyltransferase (*pflB*) (up to 6-fold) which play an important role in the biosynthesis of proteins, DNA, and RNA under anaerobic conditions. These genes exhibit markedly higher expression in biofilm cells compared to planktonic cells [45]. These enzymes contribute to formic acid production, inducing acidification in the biofilm environment. The acidification is detrimental to the host immune response, bolstering *S. aureus* persistence. It is possible that the presence of *S. aureus* in PCs might induce localized acidification, but the bacteria possess mechanisms to counter detrimental acidity. Previously, urease genes have been reported to be upregulated in biofilms in acid stress conditions [45, 55], however we observed the entire urease operon *ureABCEFGD* downregulated (S1 Table). Arginine deiminase pathway genes (*arcABCD*) and arginine biosynthetic pathway genes (*argJBCFGH*) exhibited upregulation and these pathways have been reported to facilitate pH homeostasis and biofilm maturation [56]. Anaerobic ribonucleoside reductases *nrdDG*, pivotal for DNA synthesis and repair, were notably upregulated in CBS2016-05 compared to PS/BAC/317/16/W strain. Remarkably, only PS/BAC/317/16/W displayed upregulation of aerobic ribonucleoside reductases. Genes *alsS* and *budA* were more than 5-fold upregulated in both CBS2016-05 and PS/BAC/317/16/W (**Table 1**). The *alsS* gene is known to help *S. aureus* in overcoming nitric oxide (NO) and acid stress, and resistance to antibiotics [57]. Taken together, these findings suggest that CBS2016-05 predominantly adopts a biofilm state characterized by anaerobic metabolism and acidic conditions, as reflected in the formation of small aggregates within PCs as previously reported by Loza et al (2017) [7]. The important implication for this finding is that biofilm formation has been implicated in missed bacterial detection during PC screening [7] and we speculate that growth in small anaerobic niches (biofilm aggregates) may have contributed to missed detection and subsequent transfusion of a PC contaminated with *S. aureus* CBS2016-05.

## Enhanced expression of superantigen genes in CBS2016-05 in comparison to PS/BAC/317/16/W

Staphylococcal superantigen-like (SSL) proteins SSL12, SSL13, SSL14 showed high upregulation in CBS2016-05 (up to-6 fold) and not in PS/BAC/317/16/W and these proteins are involved in immune evasion [58]. Notably, SSL10 showed 1.4-fold upregulation in CBS2016-05 and is known to be involved in inducing necroptosis by binding to the TNFR1 receptor of the human cells as recently reported by Jia et al (2022) [59] (**Table 1**). Additionally, we also observed upregulation of staphylococcal enterotoxin type H (*seh*) and an exotoxin with

unknown function in the CBS2016-05 strain. SEH is a potent inducer of cytotoxicity in T cells and has the highest affinity in low nanomolar range to bind to MHC class II molecules and has thus potential to cause cytokine storm and consequently toxic shock syndrome [60, 61]. Recently, Chi and Ramirez-Arcos, (2022b) [14] mutated the *seg* and *seh* genes, leading to enhanced growth and a significant reduction in biofilm formation in the knockout mutants when compared to the wild type strain. Also, these superantigens could be used as early biomarkers for the detection of superantigen producing *S. aureus* in PCs considering their strong involvement in septic reactions [62].

## Gene expression modulation underlying amino acid metabolism, stress resistance, and niche adaptation

A substantial downregulation (up to 5-fold) of the branched-chain amino acid (BCAA) operon in CBS2016-05 was observed. BCAAs (isoleucine, leucine, and valine) are integral to diverse protein synthesis processes, the generation of branched-chain fatty acids, and the organism's adaptability to diverse ecological niches [63]. The biosynthesis of BCAAs is intricately regulated in response to the availability of these amino acids in the environment, suggesting a potential enrichment of BCAAs within PCs. Our RNA-seq analysis further divulged the upregulation of betaine-aldehyde dehydrogenase (*betB*) and choline dehydrogenase (*betA*), pivotal enzymes driving the synthesis of betaine- an osmoprotectant metabolite crucial for cellular growth and recovery [64]. This enhanced expression underscores the strong resilience of CBS2016-05 strain against osmotic stress, reflecting an adept adaptation mechanism to thrive within challenging PC environment. Additionally, the enhanced expression of superoxide dismutase- a key enzyme orchestrating the neutralization of superoxide radicals arising from host immune responses or aerobic metabolism-further highlights the CBS2016-05 strain capacity to counteract oxidative stress [65].

Interestingly, for the CBS2016-05 strain, an upregulation of up to 7-fold was observed in the potassium transporter operon (*kdpABCDE*), with two such operons present in this strain. Conversely, in PS/BAC/317/16/W, only a solitary *kdpABCDE* operon displayed upregulation of up to 2-fold. These high-affinity, K+-specific transport systems play a pivotal role in pH homeostasis through cation transport [66]. Notably, this operon's upregulation has been documented in HEMRSA-15 (complete operon) and UAS300 (in the form of *kdpABC*) 24-hr biofilms [67]. It is to be noted that while most *S. aureus* strains carry a solitary *kdp* operon, specific strains such as MRSA252, Mu50, and N315 are reported to harbor a second *kdp* operon [68, 69]. Moreover, the KdpDE two-component system governs the expression of several virulence factors, including the positive regulation of genes related to capsular biosynthesis [70–72]. Our findings resonate with this, given the observed upregulated expression of both Cap biosynthesis and potassium transporter genes (**Table 1**). Furthermore, in clinical settings, enhanced potassium release is correlated with clotting. It is therefore plausible that PCs inoculated with *S. aureus* CBS2016-05 accumulated high potassium levels due to the upregulation of the *kdpABCDE* operon, resulting in aggregate formation, which merits further investigation.

Importantly, PCs contain linoleic acid (18:2) and oleic acid (18:1), both exhibiting antimicrobial properties against Gram-positive pathogens [73, 74]. Increased upregulation of oleate hydratase within CBS2016-05 possibly counteracts this antimicrobial immune response, effectively disarming antimicrobial unsaturated fatty acids and thereby enhancing bacterial survival within the complex milieu of PCs [75, 76].

## RT-qPCR validated the RNA-seq data

To validate the robustness of our RNA-seq findings, we conducted qRT-PCR of randomly selected eleven genes. These genes spanned diverse biological functions, encompassing capsule

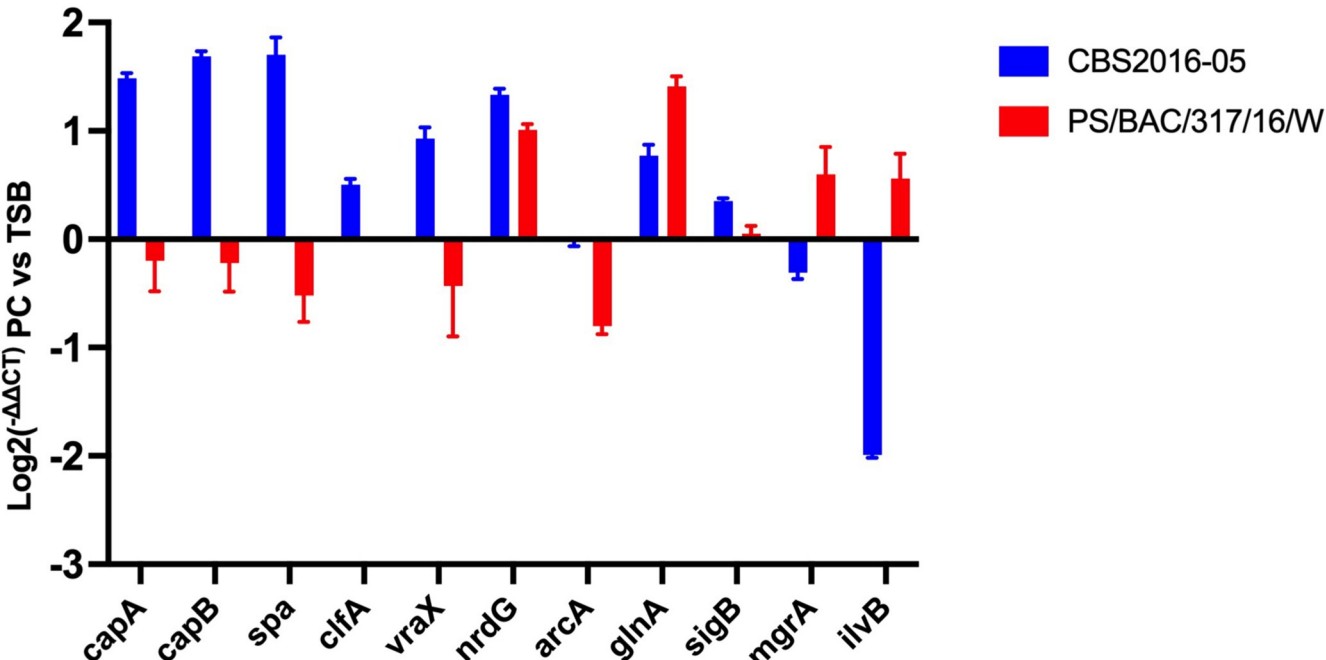

**Fig 3. RT-qPCR showing differentially expressed genes in PCs versus trypticase soy broth (TSB).** RT-qPCR of eleven randomly selected transcripts with differential gene expression (DGE) in *S. aureus* CBS2016-05 and PS/BAC/317/16/W strains in PCs versus TSB. The bars indicate the relative fold change in mRNA expression. The gyrase *gyrA* gene was employed as a normalization control.

biosynthesis (*capA* and *capB*), immune evasion and virulence (*spa*, *clfA* and *vrax*), anaerobic metabolism (*nrdG*), amino acid metabolism (*arcA*, *glnA* and *ilvA*), regulatory factors (*sigB* and *mgrA*). Through qRT-PCR gene expression analysis, the results consistently mirrored the trends observed in RNA-seq analysis (**Fig 3**), affirming the reliability and accuracy of our experimental outcomes.

## *S. aureus* induces apoptotic modulations in platelets as revealed by flow cytometry

We employed flow cytometry to assess the impact of *S. aureus* on platelet functionality. Differential expression of platelet functional and activation markers was compared between buffy coat PC units spiked with *S. aureus* CBS2016-05 and non-spiked units. The percentage of platelets positive for CD62P (P-selectin) expression was significantly increased in spiked units compared to non-spiked units at 48 hr (approximately 24.8% vs. 65.5%, $p = 0.005$) (**Fig 4A and 4B**). Notably, CD62P expression demonstrated a general increase after every 24 hr, indicating platelet activation over time during PC storage, with the presence of *S. aureus* accelerating this process. Additionally, we evaluated the expression of GP IIb and GPIbα (CD41 and CD42b), well-known platelet surface receptors for extracellular matrix molecules like fibrinogen. Spiked PCs exhibited a non-significant decrease in the percentage of GPIIb expression at 48 hr-period (approximately 97.8% vs. 71.4%, $p = 0.07$), while a significant drop was observed at 144 hr-period (approximately 96.4% vs. 49.2%, $p = 0.01$). Similarly, GPIbα expression significantly decreased at 48 hr (90.3% vs. 56.9%, $p = 0.02$) and 144 hr-period (92.6% vs. 28.6%, $p = 0.0001$) (**Fig 4A and 4B**). The impact on platelet activation was dependent on a bacterial concentration of 6E+08 CFU/mL and is consistent with our recent observation in single donor apheresis PCs [24]. Furthermore, the percentage of phosphatidylserine+ platelets significantly

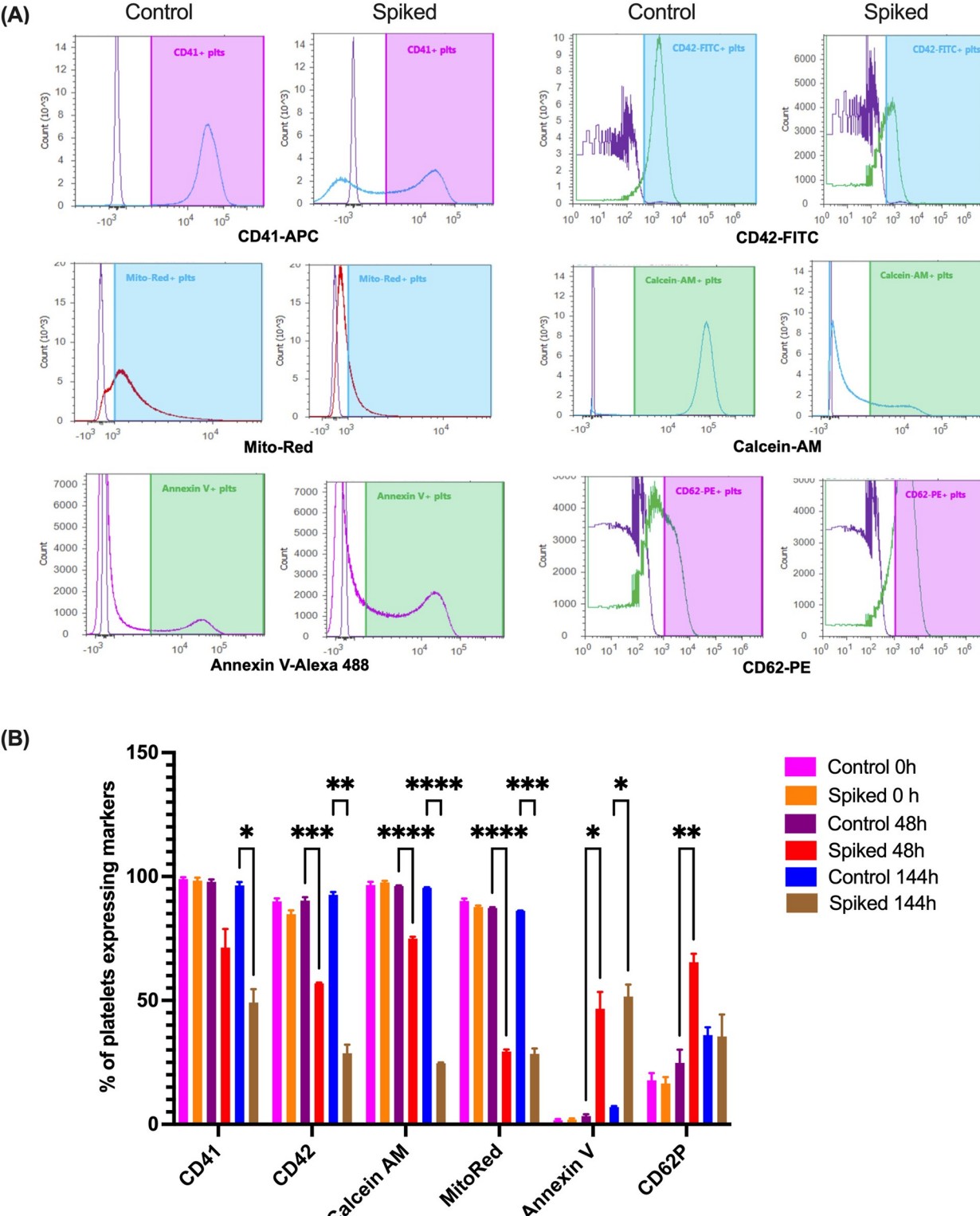

**Fig 4. *S. aureus* growth in PCs induces platelet activation and apoptosis.** PC units were spiked with *S. aureus* CBS2016-05 and incubated under standard PC condition and analyzed at 0, 24 and 144 hr. Non-spiked PC units were used as a baseline. (**A**) Representative FACS plots showing modulation in different markers used for assessment of platelet activation and functionality. (**B**) FACS analysis of platelets revealed significant decrease in expression of GPIIb (CD41, integrin αIIb), GPIbα (CD42b), whereas P-selectin CD62P) and phosphatidylserine (annexin V) showed significantly increased expression in spiked vs non-spiked platelets in a cell density dependent manner. Spiked PCs also showed significant

decrease in MitoTrack Red FM and Calcein AM positive platelets. Asterisks indicate statistical significance (* P<0.05, ** P<0.01, *** P<0.001, n = 3).

increased in spiked PCs after 48 hr of storage (3.4% vs. 46.7%, $p = 0.02$) and further increased at 144 hr (7% vs. 51.6%, $p = 0.02$), indicating apoptotic-like changes. It is therefore important to note that loss of platelet functionality with low bacterial counts is possible but not detectable in our assays which are dependent on bacterial density.

Considering the role of mitochondria in the maintenance of cellular metabolism, impairment in platelet mitochondrial function is a critical parameter that can be used to evaluate the fate of the platelets. Thus, we used mitochondrial-specific MitoTrack™ Red FM (ABP Biosciences) for evaluating the impact of bacterial contamination on platelet mitochondrial health [77], and we observed a significant decrease in mitochondrial membrane potential in spiked PCs at 48 and 144 hr of storage (87.3% vs. 29.4%, $p = 0.0001$), providing evidence of membrane depolarization. This finding was further corroborated with a Calcein AM assay, which demonstrated a significant decrease in platelet vitality and metabolic activity at 48 and 144 hr (96.1% vs. 75%, $p = 0.0005$; 95.4% vs. 24.7%, $p = 0.0001$) (**Fig 4A and 4B**). Overall, spiked PCs exhibited features consistent with apoptosis at 48 hr when the *S. aureus* concentration was 6E+08 CFU/ml, as indicated by increased annexin V expression and decreased Mito-Track Red FM and Calcein AM signals.

## Conclusion

Our comparative transcriptomic analysis between *S. aureus* CBS2016-05 and PS/BAC/317/16/W strains in two milieus, TSB and PCs, has highlighted an important repertoire of virulence factors possibly involved in platelet dysfunction and immune evasion. Specifically, an upregulation of genes encoding for capsule biosynthesis (*capA-H*), surface adhesion factors (*sasADF*), clumping factor A (*clfA*), and protein A (*spa*) within CBS2016-05 has been unveiled when grown in PCs. Upregulation of these virulence factors might have contributed to the fact that this strain escaped detection during routine PC culture screening and caused a severe transfusion reaction.

Moreover, our investigation into the interplay between *S. aureus* CBS2016-05 and PCs have demonstrated that this strain stimulates a drastic decline in GPIIb (CD41), GPIbα (CD42b), MitoTrack Red FM and Calcein AM positive platelets, accompanied by heightened P-selectin (CD62P) and phosphatidylserine (annexin V) expression. These changes are reflective of platelet activation and compromised mitochondrial functionality, highlighting that *S. aureus* can strongly impact platelet behavior and possibly introduce apoptotic behaviors.

Collectively, our findings enhanced our understanding of platelet-bacteria interactions, underscoring the enhanced pathogenicity and new roles for *S. aureus* capsule on bacterial fitness and survival in harsh environments. Differential gene expression revealed potential mechanisms for missed detection of *S. aureus* during routine PC screening, such as growth in anaerobic biofilm niches contributing to sampling error. Utilizing platelet-*S. aureus* interactions as a model system becomes highly relevant, as the consequences of these interactions are likely to play significant roles in shaping infection and host defense. Understanding bacterial modulation in PCs and their effect on platelet function provide important data to propose interventions that safeguard transfusion practices and patient well-being.

## Supporting information

**S1 Fig. Gene ontology functional enrichment analysis was performed using ShinyGO enrichment tool.** Significantly enriched pathways in *S. aureus* CBS2016-05 and PS/BAC/317/

16/W strains presented as Dot plots with a p. adjust threshold cut-off of 0.05. Enrichment significance is indicated by bubble color, while bubble size corresponds to gene count in the term.
(TIF)

**S1 Table. List pf primers used for qRT-PCR analysis of randomly selected upregulated and downregulated DEGs for S. aureus CBS2016-05 and PS/BAC/317/16/W strains when grown in PCs vs TSB.**
(XLSX)

**S2 Table. List of DEGs in S. aureus PS/Bac/317/16/W strain spiked PCs vs TSB with log2-fold-change $\geq$ 2 or $\leq$ -2, p$<$0.05.**
(XLSX)

## Acknowledgments

We thank the volunteer blood donors and staff at the Blood4Research Facility in Vancouver for whole blood collection and PC manufacturing. The authors would also like to acknowledge the assistance of the Ottawa Bioinformatics Core Facility (uOttawa/OHRI), RRID: SCR_022466. B.Y. held a post-doctoral fellowship from Canadian Blood Services during the development of this study, which was funded by Canadian Blood Services (intramural grant awarded to S. R-A.) and Health Canada. The views expressed herein do not necessarily represent the views of the federal government of Canada.

## Author Contributions

**Conceptualization:** Basit Yousuf, Sandra Ramirez-Arcos.

**Formal analysis:** Basit Yousuf, Roya Pasha.

**Funding acquisition:** Sandra Ramirez-Arcos.

**Investigation:** Basit Yousuf.

**Methodology:** Basit Yousuf, Roya Pasha, Nicolas Pineault.

**Validation:** Basit Yousuf.

**Writing – original draft:** Basit Yousuf.

**Writing – review & editing:** Basit Yousuf, Roya Pasha, Nicolas Pineault, Sandra Ramirez-Arcos.

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
