## [Decision Letter · Decision Letter 0]

12 Jun 2024

PONE-D-23-40410Modulation of Staphylococcus aureus gene expression during proliferation in platelet concentrates with focus on virulence and platelet functionalityPLOS ONE

Dear Dr. Yousuf,

Thank you for submitting your manuscript to PLOS ONE. After careful consideration, we feel that it has merit but does not fully meet PLOS ONE’s publication criteria as it currently stands. Therefore, we invite you to submit a revised version of the manuscript that addresses the points raised during the review process.

We look forward to receiving your revised manuscript.

Kind regards,

Keun Seok Seo, Ph.D.

Academic Editor

PLOS ONE

Journal Requirements:

3. Thank you for stating the following financial disclosure: "The project was funded by Canadian Blood Services (intramural grant awarded to Sandra Ramirez-Arcos) and Health Canada. " 

4. Thank you for stating the following in the Acknowledgments Section of your manuscript:"We thank the volunteer blood donors and staff at the Blood4Research Facility in Vancouver for whole blood collection and PC manufacturing. The authors would also like to acknowledge the assistance of the Ottawa Bioinformatics Core Facility (uOttawa/OHRI), RRID:SCR_022466. B.Y. held a post-doctoral fellowship from Canadian Blood Services during the development of this study, which was funded by Canadian Blood Services (intramural grant awarded to S. R-A.) and Health Canada. The views expressed herein do not necessarily represent the views of the federal government of Canada." 

Please remove any funding-related text from the manuscript and let us know how you would like to update your Funding Statement. Currently, your Funding Statement reads as follows: "The project was funded by Canadian Blood Services (intramural grant awarded to Sandra Ramirez-Arcos) and Health Canada. "

Reviewers' comments:

Reviewer's Responses to Questions

**Comments to the Author**

1. Is the manuscript technically sound, and do the data support the conclusions?

Reviewer #1: Yes

2. Has the statistical analysis been performed appropriately and rigorously? 

Reviewer #1: I Don't Know

3. Have the authors made all data underlying the findings in their manuscript fully available?

Reviewer #1: Yes

4. Is the manuscript presented in an intelligible fashion and written in standard English?

Reviewer #1: Yes

5. Review Comments to the Author

Reviewer #1: The manuscript is presented in an intelligible fashion and written in standard English. A few minor changes are suggested:

- Line 210: It would be interesting to know how platelet counts in the units correlate with flow analysis of activation and apoptosis indicators.

- It is interesting to know the K levels, PH value and external inspection of the units (presence of clots) of the tested units in correlation with the gene expression as you found for example that for the CBS2016-05 strain, an upregulation of up to 7-fold was observed in the potassium transporter operon (kdpABCDE), with two such operons present in this strain (Line 406)

- Please include limitations of this study such as the fact that functionality depends on bacterial density.

- How might this be the source of delayed detection of PC units, which are typically held in blood banks for 5-7 days. What are the likely causes of missing detection of this bacterium during routine PC screening using automated culture systems?

6. PLOS authors have the option to publish the peer review history of their article (what does this mean?). If published, this will include your full peer review and any attached files.

Reviewer #1: **Yes: **Anwar Rjoop

---

## [Author Response · Author response to Decision Letter 0]

18 Jun 2024

Response to Reviewer 1 Comments:

The manuscript is presented in an intelligible fashion and written in standard English. A few minor changes are suggested:

- Line 210: It would be interesting to know how platelet counts in the units correlate with flow analysis of activation and apoptosis indicators. 

Response: Platelet counts for activation and apoptosis assays were performed with samples containing 10-40 x 10^6 platelets/mL, which are the platelet counts required for optimal performance of these assays.

- It is interesting to know the K levels, PH value and external inspection of the units (presence of clots) of the tested units in correlation with the gene expression as you found for example that for the CBS2016-05 strain, an upregulation of up to 7-fold was observed in the potassium transporter operon (kdpABCDE), with two such operons present in this strain (Line 406) 

Response: We thank the Reviewer for this important observation. We did not determine pH or K levels in the tested units and therefore cannot directly correlate the upregulation of the kdp operon with clot formation. However, this could be one of the explanations for the aggregates observed in the inoculated units. We added the potential for this correlation in the discussion of the revised manuscript (lines 420-423). 

- Please include limitations of this study such as the fact that functionality depends on bacterial density. 

Response: This is an important consideration and we have addressed this in the revised manuscript (lines 467-468).

- How might this be the source of delayed detection of PC units, which are typically held in blood banks for 5-7 days. What are the likely causes of missing detection of this bacterium during routine PC screening using automated culture systems? 

Response: Platelet concentrates screening is done at approximately 36 hours post blood collection. Bacterial titers are typically very low (<0.01 CFU/ml) in platelet concentrates and if the isolate is a slow grower or a biofilm former, the titer could be under the limit of the detection of culture methods when sampling is done. We have added a sentence to address this question in the revised manuscript (lines 510 and 511).

Response to EDITOR Comments:

1. Please ensure that your manuscript meets PLOS ONE's style requirements https://journals.plos.org/plosone/s/file?id=wjVg/PLOSOne_formatting_sample_main_body.pdf and 

Response: The manuscript complies with PLOS ONE’ style.

2. We suggest you thoroughly copyedit your manuscript for language usage, spelling, and grammar 

Response: The manuscript has been reviewed for proper language usage, spelling and grammar.

3. Thank you for stating the following financial disclosure: "The project was funded by Canadian Blood Services (intramural grant awarded to Sandra Ramirez-Arcos) and Health Canada. " 

Please state what role the funders took in the study. If the funders had no role, please state: ""The funders had no role in study design, data collection and analy sis, decision to publish, or preparation of the manuscript."" 

Response: In cover letter, we have added the statement “The funders had no role in study design, data collection and analysis, decision to publish, or preparation of the manuscript”.

4. We note that you have included the phrase “data not shown” in your manuscript. Unfortunately, this does not meet our data sharing requirements. PLOS does not permit references to inaccessible data. 

Response: The phrase “data not shown” has been removed from the manuscript as the statement is supported by our recent paper as a reference.

Please review your reference list to ensure that it is complete and correct. 

Response: The reference list and citations in the manuscript have been reviewed.

---

## [Editor Report · Decision Letter 1]

10 Jul 2024

Modulation of Staphylococcus aureus gene expression during proliferation in platelet concentrates with focus on virulence and platelet functionality

PONE-D-23-40410R1

Dear Dr. Yousuf

We’re pleased to inform you that your manuscript has been judged scientifically suitable for publication and will be formally accepted for publication once it meets all outstanding technical requirements.

Kind regards,

Keun Seok Seo, Ph.D.

Academic Editor

PLOS ONE

Additional Editor Comments (optional):

All comments were appropriately addressed
---

## [Editor Report · Acceptance letter]

16 Jul 2024

PONE-D-23-40410R1 

PLOS ONE

Dear Dr. Yousuf, 

I'm pleased to inform you that your manuscript has been deemed suitable for publication in PLOS ONE. Congratulations! Your manuscript is now being handed over to our production team.

Kind regards, 

on behalf of

Dr. Keun Seok Seo 

Academic Editor

PLOS ONE